# Claulansine F–Donepezil Hybrids as Anti-Alzheimer’s Disease Agents with Cholinergic, Free-Radical Scavenging, and Neuroprotective Activities

**DOI:** 10.3390/molecules26051303

**Published:** 2021-02-28

**Authors:** Yingda Zang, Ke Liu, Weiping Wang, Chuangjun Li, Jie Ma, Jingzhi Yang, Xinyi Chen, Xiaoliang Wang, Dongming Zhang

**Affiliations:** State Key Laboratory of Bioactive Substance and Function of Natural Medicines, Institute of Materia Medica, Chinese Academy of Medical Sciences and Peking Union Medical College, Beijing 100050, China; zydaitfj@imm.ac.cn (Y.Z.); liuke@imm.ac.cn (K.L.); wangwp@imm.ac.cn (W.W.); lichuangjun@imm.ac.cn (C.L.); majie@imm.ac.cn (J.M.); yjzh@imm.ac.cn (J.Y.); chenxinyi@imm.ac.cn (X.C.); wangxl@imm.ac.cn (X.W.)

**Keywords:** Alzheimer’s disease, Claulansine F–donepezil hybrids, AChE inhibitory activity, free-radical scavenging activity, neuroprotective effects

## Abstract

The multifactorial nature of Alzheimer’s disease (AD) calls for the development of multitarget agents addressing key pathogenic processes. A total of 26 Claulansine F–donepezil hybrids were designed and synthesized as multitarget drugs. Among these compounds, six compounds exhibited excellent acetylcholinesterase (AChE) inhibitory activity (half maximal inhibitory concentration (IC_50_) 1.63–4.62 μM). Moreover, (*E*)-3-(8-(*tert*-Butyl)-3,3-dimethyl-3,11-dihydropyrano[3,2-a]carbazol-5-yl)-*N*-((1-(2-chlorobenzyl)piperidin-4-yl)methyl)acrylamide (**6bd**) exhibited better neuroprotective effects against OGD/R (oxygen–glucose deprivation/reoxygenation) than lead compound Claulansine F. Furthermore, **6bd** could cross the blood–brain barrier in vitro. More importantly, compared to edaravone, **6bd** had stronger free-radical scavenging activity. Molecular docking studies revealed that **6bd** could interact with the catalytic active site of AChE. All of these outstanding in vitro results indicate **6bd** as a leading structure worthy of further investigation.

## 1. Introduction

Alzheimer’s disease (AD) is the most prominent form of dementia among the population above 65 that results in a substantial burden on human health, economy, and society throughout the world. Considering the etiology of AD still remains difficult to understand, strenuous efforts are being made to elucidate the underlying mechanisms of the disease [1,2,3,4,5]. Many hypotheses including cholinesterase, β-amyloid (Aβ) deposits, tau-protein aggregation, oxidative stress, inflammation, and dyshomeostasis of biometals have been developed to illuminate the mechanism of AD [6,7]. 

At present, the treatment of AD is mainly based on cholinergic hypothesis [8]. The hypothesis emphasizes that decreased acetylcholine can lead to cognitive and memory impairment, and sustaining or recovering the cholinergic function is supposed to be clinically beneficial [9]. In addition, oxidative stress is one of the earliest events in AD pathogenesis. Compelling evidence indicates that free radicals are extremely important in causing neuronal death [10]. Antioxidants are thought to offer a good possibility of combating neurodegeneration [11,12,13]. 

*Clausena lansium*, a fruit tree distributed in southern China, is used in traditional medicine. During our continued investigation of the bioactive constituents of *Clausena lansium*, a considerable number of carbazole alkaloids were obtained and evaluated primarily for their neuroprotective effects via phytochemical studies and activity tracking [14]. Among these carbazole alkaloids, Claulansine F (Clau F), a new pyrano[3, 2-a]carbazole alkaloid, displayed remarkable protection of PC12 cells against sodium nitroprusside (SNP)-induced apoptosis [14]. Our previous pharmacological research also demonstrated that Clau F exerts a notable effect on ·OH scavenging and the protection of mitochondrial integrity [15]. With the aim of improving the activity, our research team synthesized a series of derivatives of Clau F. Subsequently, preliminary researches were conducted on the vitro neuroprotective activity of Clau F and its derivatives. Among these derivatives, CZ-7 showed the strongest ability to scavenge free radicals and penetrate the blood–brain barrier, as well as the strongest neuroprotective impact. More importantly, CZ-7 exposure in the brain was 4.3-fold higher than that in plasma, and preliminary acute toxicological test results showed that CZ-7 had no toxic side effects with an oral 1 g/kg dose in ICR (Institute of Cancer Research) mice [16]. In addition, CZ-7 has therapeutic potential for VD (vascular dementia) by alleviating oxidative stress injury through Nrf2-mediated antioxidant responses [17].

Donepezil, a Food and Drug Administration (FDA)-approved drug, is a potent, selective, uncompetitive, reversible acetylcholinesterase (AChE) inhibitor that is believed to enhance cholinergic function by increasing ACh levels in the central nervous system (CNS) [18]. However, clinical trials of this single-target drug on this disease have been modest and transient because of the multifactorial nature of AD, and this has consequently failed in the treatment of moderate to severe AD patients [19].

Considering the facts that both donepezil and Clau F exhibit a vast spectrum of biological properties and have not been combined into one drug candidate as an anti-AD agent, we hybridized the pharmacophores of donepezil and Clau F into one molecule (Figure 1). As known, the benzylpiperidine fragment of donepezil that interacts with the catalytic anionic site (CAS) of AChE has been shown as the AChE inhibitory pharmacophore. The indanone moiety can interact with the peripheral anionic site (PAS) of AChE via aromatic stacking interactions and inhibit Aβ aggregation. In order to introduce free-radical scavenging activity and diversify the structures, we replaced the indanone moiety with Clau F or its analogue CZ-7 fragments. These new derivatives may simultaneously possess dual binding sites for AChE inhibition, free-radical scavenging activity, and neuroprotective effects.

## 2. Results

### 2.1. Chemistry

The target compounds **6aa–ch** were synthesized, as outlined in Scheme 1. The key intermediates **2a,b** were prepared via the reaction between **1a,b** and dry malonic acid using piperidine as a catalyst in dry pyridine. In order to afford the other key intermediates, with an *N*-benzyl piperidine moiety **5a–n**, two-step reactions were carried out. Firstly, commercially available 4-Boc-aminopiperidine (**3a**), 4-(Boc-aminomethyl) piperidin (**3b**), and 4-(2-Boc-aminoethyl) piperidine (**3c**) were reacted with different substituted benzyl bromides to give **4a–n**. Secondly, the key intermediates **5a–n** were acquired via the reaction between trifluoroacetic acid (TFA) and **4a–n** in good yields without further purification. Lastly, **5a–n** reacted with **2a,b** in dichloromethane (DCM) catalyzed by *N*-(3-(dimethylamino)propyl)-*N*-ethylcarbodiimide hydrochloride (EDCI) and 1-hydroxybenzotriazole hydrate (HOBt) to obtain the target compounds **6aa–ch**. All final compounds (**6aa-ch**; Appendix A) were fully characterized by ^1^H and ^13^C nuclear magnetic resonance (NMR) and high-resolution electrospray ionization mass spectrometry (HRESIMS). The purity of Clau F and its synthetic analogues was determined by high-performance liquid chromatography (HPLC) analysis and was greater than 95% (Appendix A).

### 2.2. Evaluation of Cholinesterase Inhibitory Activities and Structure–Activity Relationship (SAR) Discussion

The inhibitory activities of the target compounds on AChE (from electric eel) and butyrylcholinesterase (BuChE, from equine serum) were tested using the spectrophotometric method of Ellman et al. with donepezil, parental Clau F, and CZ-7 as reference compounds. The half maximal inhibitory concentration (IC_50_) values for eeAChE and eqBuChE inhibition are shown in Table 1. Generally, the activity of cholinesterase increased after introducing the *N*-benzylpiperidine moiety mimicking donepezil compared with Clau F and CZ-7. Firstly, under the same linker condition, the AChE inhibitory activity of the compounds with R = *t*-Bu were higher than those with R = OCH_3_, for example, **6aa–ad** vs. **6ae–ah**; **6ba–bf** vs. **6bg–bj**; **6ca–cd** vs. **6ce–ch**. In particular, compounds **6ba–bf** exhibited better AChE inhibitory activity (IC_50_ 1.63–4.62 μM) than other compounds. Secondly, the length of the alkyl chain between the carbazole and donepezil moiety could significantly influence the AChE inhibitory activity. Compounds with *n* = 1 held relatively better activities than *n* = 0 or 2. In addition, the substituent *N*-benzylpiperidine moiety had no significant effect on the activity. On the other hand, compounds **6ba–bf** with R = *t*-Bu showed comparable potency (IC_50_ 5.46–19.01 μM) to donepezil (4.20 μM) against BuChE. According to the experimental results above, the compounds with R = *t*-Bu (**6aa–ad, 6ba–bf,** and **6ca–cd**) were selected as the candidate compounds for subsequent investigations.

### 2.3. Neuroprotective Effects against OGD/R (Oxygen–Glucose Deprivation/Reoxygenation) Injury on Primary Cortical Neurons

The parental compounds Clau F and CZ-7 have remarkable neuroprotective effects. To further confirm whether the derivatives retain activity, an OGD/R model was performed in primary cortical neurons, and 3-(4,5-dimethylthiazol-2-yl)-2,5-diphenyltetrazolium bromide (MTT) assays were used to test cell viability. As shown in Figure 2, OGD/R treatment obviously reduced the cell viability of cultured neurons to 40.4% from 100% in the control group. The administration of most of the target compounds (10 μM) resulted in elevated levels of cell viability. In particular, **6bb**, **6bd**, **6be**, **6bf**, and parental comopound CZ-7 significantly promoted cell viability to 109%, 125.6%, 104.7%, 107.3%, and 112.7%, respectively (*n* = 3, *p* < 0.01 vs. OGD/R group). According to the experimental results above, compound **6bd** was selected as the candidate compound for the subsequent investigations.

### 2.4. Brain Penetration

A blood–brain barrier (BBB) penetration assay in vitro is essential for anti-AD drug development. A parallel artificial membrane permeation assay for BBB (PAMPA-BBB) was carried out to investigate the BBB permeability of compound **6bd**. According to the obtained data on permeability in Table 2, **6bd** was classified among the compounds having good brain penetration with a Pe value of 4.38 ± 0.05 × 10^−6^ cm·s^−1^ (Pe represents the PAMPA effective permeability coefficient). The test was performed using a good (corticosterone) and weak brain-penetrating reference (ofloxacin), respectively.

### 2.5. Free-Radical Scavenging Ability of ***6bd***

Oxidative stress caused by reactive oxygen (for example, singlet oxygen and hydroxyl radical) is related to the death of neurons and the formation of intracellular Aβ oligomer [21,22]. Drugs designed to prevent the formation or removal of free radicals in the brain are beneficial for AD. The main technique for detecting organic free radicals is electron spin resonance (ESR) spectroscopy. This technology can detect radical concentrations on the order of 10 nM or less under favorable conditions and has high sensitivity. Therefore, compound **6bd** was tested for its ability to scavenge active oxygen by using ESR spectroscopy. For comparison, edaravone (Eda) was used as a reference. The singlet oxygen scavenging of 10^−2^ mol/L compound **6bd** or Eda is shown in Figure 3, with the former having a stronger effect than the latter.

### 2.6. Molecular Docking of ***6bd*** on AChE

To further investigate the binding modes of these derivatives, a molecular docking study was performed with the most promising compound 6bd using the Discovery Studio software package. In this study, the X-ray crystal structure of the *h*AChE complex with donepezil (Protein Data Bank identifier (PDB ID): 4EY7) was applied to this study [23]. Redocking simulation was performed with the native ligands to verify the docking procedures applied. The site of the grid box was on the centroid of the pre-dock ligand with X-ray coordinates of X: −13.983, Y: 43.975, and Z: 27.900. As a result, donepezil highly overlapped with the co-crystal ligand (root-mean-square deviation (RMSD) value = 0.18 Å). After validating the docking method, compound **6bd** was docked using the Discovery Studio software package. As illustrated in Figure 4, the predicted binding mode of compound **6bd** was very similar to the experimental conformation observed for the AChE–donepezil complex, and they retained several key interactions. The phenyl moiety of **6bd** was oriented toward the CAS of AChE through the π-stacking interaction with the phenyl ring of Trp86. The charged nitrogen of piperidine ring could bind to the CAS by making a cation–π interaction with the aromatic ring in Tyr337 [24]. A π–alkyl interaction was observed between the F atom of compound **6bd** and His447. Furthermore, **6bd** formed a hydrogen bond with Asp74 and Try341 of the PAS. Meanwhile, the carbazole ring of compound 6bd formed a π–π interaction with Trp286. All these results clearly indicated that compound 6bd was suitable for the active site of the enzyme, where it simultaneously interacted with the PAS and CAS of *h*AChE.

## 3. Discussion

### 3.1. Chemistry

All commercial chemicals were used as supplied unless otherwise indicated. All reactions were performed in oven-dried glassware. All yields reported refer to the yields of the isolated compounds. ^1^H-NMR and ^13^C-NMR spectral data were obtained using Bruker Avance 600 MHz and 400 MHz spectrometers at 300 K using tetramethylsilane (TMS) as an internal standard. HRESIMS was measured with an Agilent 6520 Accurate-Mass Q-TOF LC/MS. Silica gel thin-layer chromatography (TLC) plates (Qing Dao Marine Chemical Factory, Qingdao, China) were used to monitor the reaction progress. Column chromatography was performed using silica gel (100–200 mesh size, Qing Dao Marine Chemical Factory, Qingdao, China). Compounds **5a–n** were synthesized as previously described [25].

#### 3.1.1. General Procedure for the Synthesis of **2a,b**

A solution of **1a,b** (1 equivalent (eq.)) and malonic acid (3 eq.) in a mixture of pyridine (10 mL·mmol^−1^
**1a,b**) and piperidine (0.01 eq.) was maintained at 40 °C for 48 h. The solution was concentrated by evaporation, the residue was diluted with 250 mL of water, and the pH was adjusted to 11.0 with 2 N NaOH. The solution was extracted with four 200 mL portions of ethyl acetate to remove bases and neutral impurities. The aqueous phase was treated with charcoal, and the filtrate was acidified to pH 1.8 by dropwise addition of 6 N hydrochloric acid with stirring at 5 °C. The resulting yellow precipitate was washed well with cold water to obtain **2a,b.**

*(E)-3-(8-Methoxy-3,3-dimethyl-3,11-dihydropyrano[3,2-a]carbazol-5-yl)acrylic acid***(2a).** White solid, 75%; melting point (m.p.): 182–183 °C. ^1^H NMR (400 MHz, dimethyl sulfoxide (DMSO)-*d*_6_) δ 12.09 (s, 1H), 11.35 (s, 1H), 8.47 (s, 1H), 8.13 (d, *J* = 1.9 Hz, 1H), 7.96 (d, *J* = 16.0 Hz, 1H), 7.40 (dd, *J* = 8.5, 1.9 Hz, 1H), 7.34 (d, *J* = 8.5 Hz, 1H), 6.92 (d, *J* = 9.8 Hz, 1H), 6.62 (d, *J* = 16.0 Hz, 1H), 5.87 (d, *J* = 9.8 Hz, 1H), 1.48 (s, 6H), 1.39 (s, 9H). ^13^C-NMR (100 MHz, DMSO-*d*_6_) δ 168.4, 149.5, 142.0, 139.7, 138.5, 129.3, 122.9, 122.8, 119.6, 117.7, 117.4, 116.3, 115.5, 114.4, 110.3, 104.2, 76.6, 34.4, 31.8, 27.4. HRESIMS *m/z* = 376.19064 [M + H]^+^ (calculated for C_24_H_26_O_3_N, 376.19072).

*(E)-3-(8-(tert-Butyl)-3,3-dimethyl-3,11-dihydropyrano[3,2-a]carbazol-5-yl)acrylic acid***(2b).** White solid, 68%; m.p.: 205–207 °C. ^1^H-NMR (400 MHz, DMSO-*d*_6_) δ 12.11 (s, 1H), 11.30 (s, 1H), 8.41 (s, 1H), 7.95 (d, *J* = 16.0 Hz, 1H), 7.70 (d, *J* = 2.5 Hz, 1H), 7.33 (d, *J* = 8.7 Hz, 1H), 6.96 (dd, *J* = 8.7, 2.5 Hz, 1H), 6.90 (d, *J* = 9.8 Hz, 1H), 6.58 (d, *J* = 16.0 Hz, 1H), 5.86 (d, *J* = 9.9 Hz, 1H), 3.83 (s, 3H), 1.48 (s, 6H). ^13^C-NMR (100 MHz, DMSO-*d*_6_) δ 168.4, 153.6, 149.7, 139.7, 138.7, 135.0, 129.3, 123.6, 119.9, 117.6, 117.3, 115.5, 114.3, 114.0, 111.5, 104.3, 103.2, 76.7, 55.5, 27.4. HRESIMS *m/z* = 350.13959 [M + H]^+^ (calculated for C_21_H_20_O_4_N, 350.13868).

#### 3.1.2. General Procedure for the Synthesis of **6aa–ch**

To a solution of **2a,b** (1 eq.) and **5a–n** (2 eq.) in dry tetrahydrofuran (THF, 10 mL·mmol^−1^
**2a,b**), EDCI (2.0 eq.) and HOBt (2.0 eq.) were added by stirring at room temperature for 24 h. The solvent was removed by evaporation, and the residue was diluted with EtOAc. The solution was washed with water, dried over MgSO_4_, filtered, and concentrated in vacuo. Purification by column chromatography (petroleum ether/ethyl acetate, 3:1) gave compounds **6aa–ch.**

*(E)-N-(1-Benzylpiperidin-4-yl)-3-(8-(tert-butyl)-3,3-dimethyl-3,11-dihydropyrano[3,2-a]carbazol-5-yl)acrylamide***(6aa)**. White solid, 58%; m.p.: 191–192 °C. ^1^H-NMR (400 MHz, DMSO-*d*_6_) δ 11.31 (s, 1H), 8.16 (s, 1H), 8.01 (s, 1H), 7.93 (d, *J* = 7.7 Hz, 1H), 7.74 (d, *J* = 15.7 Hz, 1H), 7.44–7.22 (m, 7H), 6.92 (d, *J* = 9.8 Hz, 1H), 6.72 (d, *J* = 15.8 Hz, 1H), 5.86 (d, *J* = 9.8 Hz, 1H), 3.76–3.61 (m, 1H), 3.48 (s, 2H), 2.84–2.74 (m, 2H), 2.17–2.01 (m, 2H), 1.83–1.75 (m, 2H), 1.48 (s, 6H), 1.46–1.43 (m, 2H), 1.38 (s, 9H). ^13^C-NMR (100 MHz, DMSO-*d*_6_) δ 165.0, 149.3, 141.8, 138.4, 138.4, 137.8, 134.5, 129.4, 128.8, 128.2, 126.9, 122.8, 122.6, 119.7, 118.7, 117.4, 117.4, 115.6, 115.4, 110.3, 104.4, 76.5, 62.1, 51.9, 45.9, 34.4, 31.8, 31.7, 27.4. HRESIMS *m/z* = 584.32721 [M + H]^+^ (calcd for C_36_H_42_O_2_N_3_, 548.32715).

*(E)-3-(8-(tert-Butyl)-3,3-dimethyl-3,11-dihydropyrano[3,2-a]carbazol-5-yl)-N-(1-(2-fluorobenzyl)piperidin-4-yl)acrylamide***(6ab).** White solid, 56%; m.p.: 168–169 °C. ^1^H NMR (400 MHz, DMSO-*d*_6_) δ 11.34 (s, 1H), 8.16 (s, 1H), 8.00 (s, 2H), 7.75 (d, *J* = 15.7 Hz, 1H), 7.54–7.30 (m, 4H), 7.28–7.13 (m, 2H), 6.93 (d, *J* = 9.8 Hz, 1H), 6.72 (d, *J* = 15.7 Hz, 1H), 5.85 (d, *J* = 9.8 Hz, 1H), 3.88–3.51 (m, 3H), 3.07–2.77 (m, 2H), 2.50 (s, 2H), 1.93–1.78 (m, 2H), 1.62–1.50 (m, 2H), 1.48 (s, 6H), 1.38 (s, 9H). ^13^C NMR (100 MHz, DMSO-*d*_6_) δ 165.1, 160.9 (d, *J*_C-F_ = 245.8 Hz), 149.3, 141.8, 138.4, 137.8, 134.6, 132.0 (d, *J*_C-F_ = 4.7 Hz), 129.4, 128.7 (d, *J*_C-F_ = 8.2 Hz), 124.5 (d, *J*_C-F_ = 14.6 Hz), 124.4 (d, *J*_C-F_ = 3.4 Hz), 122.8, 122.6, 119.6, 118.6, 117.5, 117.4, 115.6, 115.3, 115.2 (d, *J*_C-F_ = 22.1 Hz), 110.4, 104.4, 76.5, 59.7, 51.5, 45.0, 34.4, 31.8, 31.7, 27.4. HRESIMS *m*/*z* = 566.31696 [M + H]^+^ (calculated for C_36_H_41_O_2_N_3_F, 566.31773).

*(E)-3-(8-(tert-Butyl)-3,3-dimethyl-3,11-dihydropyrano[3,2-a]carbazol-5-yl)-N-(1-(2-methylbenzyl)piperidin-4-yl)acrylamide***(6ac).** White solid, 52%; m.p.: 201–202 °C. ^1^H-NMR (400 MHz, DMSO-*d*_6_) δ 11.37 (s, 1H), 8.16 (s, 1H), 8.00 (d, *J* = 1.8 Hz, 1H), 7.92 (d, *J* = 7.8 Hz, 1H), 7.74 (d, *J* = 15.8 Hz, 1H), 7.40 (dd, *J* = 8.4, 1.8 Hz, 1H), 7.35 (d, *J* = 8.4 Hz, 1H), 7.23–7.20 (m, 1H), 7.16–7.11 (m, 3H), 6.93 (d, *J* = 9.8 Hz, 1H), 6.72 (d, *J* = 15.8 Hz, 1H), 5.85 (d, *J* = 9.8 Hz, 1H), 3.77–3.64 (m, 1H), 3.41 (s, 2H), 2.78–2.74 (m, 2H), 2.32 (s, 3H), 2.13–2.02 (m, 2H), 1.85–1.73 (m, 2H), 1.48 (s, 6H), 1.47–1.41 (m, 2H), 1.38 (s, 9H). ^13^C-NMR (100 MHz, DMSO-*d*_6_) δ 165.0, 149.3, 141.7, 138.4, 137.8, 137.0, 136.7, 134.5, 130.0, 129.5, 129.3, 126.8, 125.4, 122.8, 122.6, 119.8, 118.7, 117.5, 117.4, 115.6, 115.4, 110.4, 104.4, 76.5, 60.3, 52.1, 46.0, 34.4, 31.9, 31.8, 27.4, 18.8. HRESIMS *m/z* = 562.34235 [M + H]^+^ (calculated for C_37_H_44_O_2_N_3_, 562.34280).

*(E)-3-(8-(tert-Butyl)-3,3-dimethyl-3,11-dihydropyrano[3,2-a]carbazol-5-yl)-N-(1-(2-chlorobenzyl)piperidin-4-yl)acrylamide***(6ad).** White solid, 62%; m.p.: 188–189 °C. ^1^H-NMR (400 MHz, DMSO-*d*_6_) δ 11.34 (s, 1H), 8.17 (s, 1H), 8.01 (d, *J* = 1.8 Hz, 1H), 7.94 (d, *J* = 7.7 Hz, 1H), 7.76 (d, *J* = 15.8 Hz, 1H), 7.48 (dd, *J* = 7.5, 1.9 Hz, 1H), 7.43–7.25 (m, 5H), 6.93 (d, *J* = 9.8 Hz, 1H), 6.74 (d, *J* = 15.8 Hz, 1H), 5.85 (d, *J* = 9.8 Hz, 1H), 3.79–3.66 (m, 1H), 3.55 (s, 2H), 2.82–2.78 (m, 2H), 2.21–2.10 (m, 2H), 1.85–1.75 (m, 2H), 1.55–1.43 (m, 8H), 1.38 (s, 9H). ^13^C-NMR (100 MHz, DMSO-*d*_6_) δ 165.1, 149.3, 141.8, 138.4, 137.8, 136.0, 134.5, 133.2, 130.7, 129.3, 129.2, 128.5, 127.0, 122.8, 122.6, 119.7, 118.7, 117.5, 117.4, 115.6, 115.4, 110.3, 104.4, 76.5, 58.7, 52.0, 45.8, 34.4, 31.8, 31.8, 27.4.HRESIMS *m/z* = 582.28790 [M + H]^+^ (calculated for C_36_H_41_O_2_N_3_Cl, 582.28818).

*(E)-N-(1-Benzylpiperidin-4-yl)-3-(8-methoxy-3,3-dimethyl-3,11-dihydropyrano[3,2-a]carbazol-5-yl)acrylamide***(6ae).** White solid, 57%; m.p.: 212–213 °C. ^1^H-NMR (600 MHz, DMSO-*d*_6_) δ 11.26 (s, 1H), 8.13 (s, 1H), 7.94 (d, *J* = 7.4 Hz, 1H), 7.73 (d, *J* = 15.8 Hz, 1H), 7.58 (d, *J* = 2.5 Hz, 1H), 7.33 (d, *J* = 8.6 Hz, 1H), 7.33–7.27 (m, 4H), 7.27–7.21 (m, 1H), 6.96 (dd, *J* = 8.6, 2.5 Hz, 1H), 6.90 (d, *J* = 9.8 Hz, 1H), 6.70 (d, *J* = 15.8 Hz, 1H), 5.85 (d, *J* = 9.8 Hz, 1H), 3.83 (s, 3H), 3.73–3.65 (m, 1H), 3.35 (s, 2H), 2.81–2.74 (m, 2H), 2.11–2.01 (m, 2H), 1.83–1.72 (m, 2H), 1.48 (s, 6H), 1.47–1.41 (m, 2H). ^13^C-NMR (150 MHz, DMSO-*d*_6_) δ 165.1, 162.3, 153.5, 149.5, 138.6, 137.9, 134.8, 134.7, 129.3, 128.7, 128.2, 126.8, 123.4, 119.8, 119.2, 117.1, 115.4, 113.8, 111.5, 104.4, 102.8, 76.5, 62.2, 55.5, 52.0, 46.7, 31.8, 27.4. HRESIMS *m/z* = 522.27576 [M + H]^+^ (calculated for C_33_H_36_O_3_N_3_, 522.27512).

*(E)-N-(1-(2-Fluorobenzyl)piperidin-4-yl)-3-(8-methoxy-3,3-dimethyl-3,11-dihydropyrano[3,2-a]carbazol-5-yl)acrylamide***(6af).** White solid, 65%; m.p.: 195–196 °C. ^1^H-NMR (600 MHz, DMSO-d6) δ 8.13 (s, 1H), 7.94 (d, *J* = 7.2 Hz, 1H), 7.74 (d, *J* = 15.7 Hz, 1H), 7.58 (d, *J* = 2.5 Hz, 1H), 7.42–7.38 (m, 1H), 7.33 (d, *J* = 8.7 Hz, 1H), 7.32–7.28 (m, 1H), 7.20–7.13 (m, 2H), 6.96 (dd, J = 8.7, 2.5 Hz, 1H), 6.90 (d, *J* = 9.7 Hz, 1H), 6.70 (d, *J* = 15.8 Hz, 1H), 5.84 (d, *J* = 9.8 Hz, 1H), 3.83 (s, 3H), 3.68 (d, *J* = 7.2 Hz, 1H), 3.52 (s, 2H), 2.79 (d, *J* = 11.2 Hz, 2H), 2.10 (t, *J* = 11.2 Hz, 2H), 1.79 (dd, *J* = 12.8, 4.0 Hz, 2H), 1.48 (s, 6H), 1.46–1.42 (m, 2H). ^13^C-NMR (150 MHz, DMSO-d6) δ 165.1, 160.8 (d, *J*_C-F_ = 244.4 Hz), 153.5, 149.5, 137.9, 134.8, 134.7, 131.5 (d, *J*_C-F_ = 4.9 Hz), 129.3, 129.0 (d, *J*_C-F_ = 8.2 Hz), 124.9 (d, *J*_C-F_ = 14.6 Hz), 124.2 (d, *J*_C-F_ = 3.3 Hz), 123.4, 119.8, 119.2, 117.4, 117.2, 115.4, 115.1 (d, *J*_C-F_ = 21.9 Hz), 113.8, 111.5, 104.4, 102.8, 76.5, 59.8, 55.5, 51.8, 45.9, 31.7, 27.4. HRESIMS *m*/*z* = 540.26569 [M + H]+ (calculated for C_33_H_35_O_3_N_3_F, 540.26570).

*(E)-3-(8-Methoxy-3,3-dimethyl-3,11-dihydropyrano[3,2-a]carbazol-5-yl)-N-(1-(2-methylbenzyl)piperidin-4-yl)acrylamide***(6ag).** White solid, 58%; m.p.: 215–216 °C. ^1^H-NMR (600 MHz, DMSO-*d*_6_) δ 8.12 (s, 1H), 7.93 (d, *J* = 7.7 Hz, 1H), 7.72 (d, *J* = 15.8 Hz, 1H), 7.57 (d, *J* = 2.5 Hz, 1H), 7.33 (d, *J* = 8.7 Hz, 1H), 7.24–7.20 (m, 1H), 7.17–7.11 (m, 3H), 6.96 (dd, *J* = 8.7, 2.5 Hz, 1H), 6.90 (d, *J* = 9.8 Hz, 1H), 6.70 (d, *J* = 15.8 Hz, 1H), 5.85 (d, *J* = 9.8 Hz, 1H), 3.83 (s, 3H), 3.75–3.64 (m, 1H), 3.42 (s, 2H), 2.81–2.73 (m, 2H), 2.32 (s, 3H), 2.09 (s, 2H), 1.82–1.74 (m, 2H), 1.48 (s, 6H), 1.46–1.41 (m, 2H). ^13^C-NMR (150 MHz, DMSO-*d*_6_) δ 165.1, 153.5, 149.5, 137.9, 137.0, 136.7, 134.8, 134.7, 130.1, 129.5, 129.3, 126.9, 125.4, 123.4, 119.8, 119.3, 117.4, 117.1, 115.4, 113.8, 111.5, 104.4, 102.8, 76.5, 59.8, 55.5, 52.1, 46.0, 31.8, 27.4, 18.8. HRESIMS *m*/*z* = 536.29102 [M + H]+ (calculated for C_34_H_38_O_3_N_3_, 536.29007).

*(E)-N-(1-(2-Chlorobenzyl)piperidin-4-yl)-3-(8-methoxy-3,3-dimethyl-3,11-dihydropyrano[3,2-a]carbazol-5-yl)acrylamide***(6ah).** White solid, 68%; m.p.: 197–198 °C. ^1^H-NMR (600 MHz, DMSO-*d*_6_) δ 11.25 (s, 1H), 8.13 (s, 1H), 7.95 (d, *J* = 7.7 Hz, 1H), 7.74 (d, *J* = 15.8 Hz, 1H), 7.58 (d, *J* = 2.5 Hz, 1H), 7.48 (d, *J* = 7.7 Hz, 1H), 7.42 (d, *J* = 7.7 Hz, 1H), 7.36–7.25 (m, 3H), 6.96 (dd, *J* = 8.7, 2.5 Hz, 1H), 6.90 (d, *J* = 9.8 Hz, 1H), 6.72 (d, *J* = 15.8 Hz, 1H), 5.85 (d, *J* = 9.8 Hz, 1H), 3.83 (s, 3H), 3.78–3.67 (m, 1H), 3.56 (s, 2H), 2.85–2.76 (m, 2H), 2.22–2.11 (m, 2H), 1.84–1.76 (m, 2H), 1.54–1.43 (m, 8H). ^13^C-NMR (150 MHz, DMSO-*d*_6_) δ 165.1, 153.5, 149.5, 137.9, 136.0, 134.8, 134.7, 133.3, 130.7, 129.3, 129.2, 128.5, 127.0, 123.4, 119.8, 119.2, 117.4, 117.1, 115.4, 113.8, 111.5, 104.4, 102.8, 76.5, 58.7, 55.5, 52.1, 45.9, 31.8, 27.4. HRESIMS *m/z* = 556.23633 [M + H]^+^ (calculated for C_33_H_35_O_3_N_3_Cl, 556.23615).

*(E)-N-((1-Benzylpiperidin-4-yl)methyl)-3-(8-(tert-butyl)-3,3-dimethyl-3,11-dihydropyrano[3,2-a]carbazol-5-yl)acrylamide***(6ba).** White solid, 54%; m.p.: 163–164 °C. ^1^H-NMR (400 MHz, DMSO-*d*_6_) δ 11.30 (s, 1H), 8.16 (s, 1H), 8.01 (d, *J* = 1.8 Hz, 1H), 7.98 (d, *J* = 5.9 Hz, 1H), 7.74 (d, *J* = 15.8 Hz, 1H), 7.40 (dd, *J* = 8.6, 1.8 Hz, 1H), 7.35 (d, *J* = 8.6 Hz, 1H), 7.33–7.26 (m, 4H), 7.26–7.21 (m, 1H), 6.91 (d, *J* = 9.8 Hz, 1H), 6.75 (d, *J* = 15.8 Hz, 1H), 5.86 (d, *J* = 9.8 Hz, 1H), 3.45 (s, 2H), 3.12–3.06 (m, 2H), 2.86–2.77 (m, 2H), 1.91 (s, 2H), 1.69–1.59 (m, 2H), 1.48 (s, 6H), 1.40–1.36 (m, 10H), 1.28–1.12 (m, 2H). ^13^C-NMR (100 MHz, DMSO-*d*_6_) δ 165.9, 149.3, 141.8, 138.7, 138.4, 137.8, 134.4, 129.4, 128.8, 128.1, 126.9, 122.8, 122.6, 119.6, 118.6, 117.4, 117.4, 115.7, 115.4, 110.3, 104.4, 76.5, 62.4, 53.0, 44.3, 35.9, 34.4, 31.8, 29.7, 27.4. HRESIMS *m/z* = 562.34210 [M + H]^+^ (calculated for C_37_H_44_O_2_N_3_, 562.34280).

*(E)-3-(8-(tert-Butyl)-3,3-dimethyl-3,11-dihydropyrano[3,2-a]carbazol-5-yl)-N-((1-(2-fluorobenzyl)piperidin-4-yl)methyl)acrylamide***(6bb).** White solid, 52%; m.p.: 142–143 °C. ^1^H-NMR (400 MHz, DMSO-*d*_6_) δ 11.32 (s, 1H), 8.16 (s, 1H), 8.05–7.94 (m, 2H), 7.74 (d, *J* = 15.7 Hz, 1H), 7.45–7.09 (m, 6H), 6.92 (d, *J* = 9.8 Hz, 1H), 6.75 (d, *J* = 15.7 Hz, 1H), 5.85 (d, *J* = 9.8 Hz, 1H), 3.49 (s, 2H), 3.12–3.03 (m, 2H), 2.81 (d, *J* = 11.3 Hz, 2H), 1.94 (t, *J* = 11.3 Hz, 2H), 1.69–1.58 (m, 2H), 1.48 (s, 6H), 1.40–1.35 (m, 10H), 1.26–1.11 (m, 2H). ^13^C-NMR (100 MHz, DMSO-*d*_6_) δ 165.9, 160.7 (d, *J*_C-F_ = 244.1 Hz), 149.3, 141.8, 138.4, 137.8, 134.4, 131.4 (d, *J*_C-F_ = 4.7 Hz), 129.4, 128.9(d, *J*_C-F_ = 8.2 Hz), 125.0 (d, *J*_C-F_ = 14.5 Hz), 124.1(d, *J*_C-F_ = 3.4 Hz), 122.8, 122.6, 119.6, 118.6, 117.5, 117.3, 115.6, 115.4, 115.1 (d, *J*_C-F_ = 22.0 Hz), 110.3, 104.4, 76.5, 54.9, 52.9, 44.2, 35.8, 34.4, 31.8, 29.7, 27.4. HRESIMS *m*/*z* = 580.33258 [M + H]^+^ (calculated for C_37_H_43_O_2_N_3_F, 580.233338).

*(E)-3-(8-(tert-Butyl)-3,3-dimethyl-3,11-dihydropyrano[3,2-a]carbazol-5-yl)-N-((1-(2-methylbenzyl)piperidin-4-yl)methyl)acrylamide***(6bc).** White solid, 54%; m.p.: 168–169 °C. ^1^H-NMR (400 MHz, DMSO-*d*_6_) δ 11.33 (s, 1H), 8.16 (s, 1H), 8.05–7.96 (m, 2H), 7.74 (d, *J* = 15.8 Hz, 1H), 7.46–7.32 (m, 2H), 7.27–7.07 (m, 4H), 6.92 (d, *J* = 9.8 Hz, 1H), 6.75 (d, *J* = 15.8 Hz, 1H), 5.85 (d, *J* = 9.8 Hz, 1H), 3.38 (s, 2H), 3.13–3.05 (m, 2H), 2.79 (d, *J* = 11.4 Hz, 2H), 2.30 (s, 3H), 1.93 (t, *J* = 11.4 Hz, 2H), 1.70–1.59 (m, 2H), 1.48 (s, 6H), 1.48–1.41 (m, 1H), 1.38 (s, 9H), 1.25–1.06 (m, 2H). ^13^C-NMR (151 MHz, DMSO) δ 165.9, 153.5, 149.5, 137.9, 136.9, 136.7, 134.8, 134.6, 130.0, 129.4, 129.3, 126.7, 125.3, 123.4, 119.6, 119.2, 117.4, 117.1, 115.3, 113.8, 111.5, 104.4, 102.8, 76.5, 60.5, 55.5, 53.2, 44.3, 36.0, 29.9, 27.4, 18.8. HRESIMS *m*/*z* = 576.35785 [M + H]^+^ (calculated for C_38_H_46_O_2_N_3_, 576.35785).

*(E)-3-(8-(tert-Butyl)-3,3-dimethyl-3,11-dihydropyrano[3,2-a]carbazol-5-yl)-N-((1-(2-chlorobenzyl)piperidin-4-yl)methyl)acrylamide***(6bd)**. White solid, 66%; m.p.: 175–176 °C. ^1^H-NMR (400 MHz, DMSO-*d*_6_) δ 11.31 (s, 1H), 8.16 (s, 1H), 8.05–7.95 (m, 2H), 7.74 (d, *J* = 15.7 Hz, 1H), 7.51–7.23 (m, 6H), 6.92 (d, *J* = 9.8 Hz, 1H), 6.75 (d, *J* = 15.7 Hz, 1H), 5.86 (d, *J* = 9.8 Hz, 1H), 3.52 (s, 2H), 3.13–3.04 (m, 2H), 2.83 (d, *J* = 11.3 Hz, 2H), 2.01 (t, *J* = 11.3 Hz, 2H), 1.71–1.62 (m, 2H), 1.48 (s, 6H), 1.40–1.35 (m, 10H), 1.26–1.15 (m, 2H). ^13^C-NMR (101 MHz, DMSO) δ 165.9, 149.3, 141.8, 138.4, 137.8, 136.1, 134.4, 133.2, 130.6, 129.4, 129.2, 128.4, 127.0, 122.8, 122.6, 119.6, 118.6, 117.4, 117.4, 115.7, 115.4, 110.3, 104.4, 76.5, 58.9, 53.1, 44.3, 35.9, 34.4, 31.8, 29.8, 27.4. HRESIMS *m*/*z* = 596.30298 [M + H]^+^ (calculated for C_37_H_43_O_2_N_3_Cl, 596.30383).

*(E)-3-(8-(tert-Butyl)-3,3-dimethyl-3,11-dihydropyrano[3,2-a]carbazol-5-yl)-N-((1-(3-chlorobenzyl)piperidin-4-yl)methyl)acrylamide***(6be).** White solid, 59%; m.p.: 171–172 °C. ^1^H-NMR (400 MHz, DMSO-*d*_6_) δ 11.30 (s, 1H), 8.16 (s, 1H), 8.07–7.93 (m, 2H), 7.74 (d, *J* = 15.7 Hz, 1H), 7.40 (dd, *J* = 8.5, 1.9 Hz, 1H), 7.37–7.32 (m, 3H), 7.32–7.26 (m, 2H), 6.91 (d, *J* = 9.8 Hz, 1H), 6.75 (d, *J* = 15.7 Hz, 1H), 5.86 (d, *J* = 9.8 Hz, 1H), 3.46 (s, 2H), 3.14–3.05 (m, 2H), 2.83–2.74 (m, 2H), 1.97–1.85 (m, 2H), 1.69–1.61 (m, 2H), 1.48 (s, 6H), 1.38 (s, 10H), 1.25–1.16 (m, 2H). ^13^C-NMR (100 MHz, DMSO-d_6_) δ 165.9, 149.3, 141.8, 138.4, 137.8, 136.1, 134.4, 132.9, 130.0, 129.4, 128.2, 127.3, 126.8, 122.8, 122.6, 119.6, 118.6, 117.4, 117.4, 115.7, 115.4, 110.3, 104.4, 76.5, 61.5, 53.0, 44.3, 35.9, 34.4, 31.8, 29.8, 27.4. HRESIMS *m*/*z* = 596.30365 [M + H]^+^ (calculated for C_37_H_43_O_2_N_3_Cl, 596.30383).

*(E)-3-(8-(tert-Butyl)-3,3-dimethyl-3,11-dihydropyrano[3,2-a]carbazol-5-yl)-N-((1-(4-chlorobenzyl)piperidin-4-yl)methyl)acrylamide***(6bf).** White solid, 62%; m.p.: 158–159 °C. ^1^H-NMR (400 MHz, DMSO-*d*_6_) δ 11.30 (s, 1H), 8.16 (s, 1H), 8.01 (d, *J* = 1.8 Hz, 1H), 7.98 (d, *J* = 5.9 Hz, 1H), 7.74 (d, *J* = 15.8 Hz, 1H), 7.40 (dd, *J* = 8.5, 1.9 Hz, 1H), 7.40–7.32 (m, 3H), 7.31 (d, *J* = 8.5 Hz, 2H), 6.92 (d, *J* = 9.8 Hz, 1H), 6.75 (d, *J* = 15.8 Hz, 1H), 5.86 (d, *J* = 9.8 Hz, 1H), 3.42 (s, 2H), 3.13–3.04 (m, 2H), 2.82–2.72 (m, 2H), 1.96–1.84 (m, 2H), 1.68–1.58 (m, 2H), 1.48 (s, 6H), 1.38 (s, 9H), 1.26–1.11 (m, 3H). ^13^C-NMR (100 MHz, DMSO-d_6_) δ 165.9, 149.3, 141.8, 138.4, 137.8, 134.4, 131.2, 130.4, 129.4, 128.1, 122.8, 122.6, 119.6, 118.6, 117.4, 117.4, 115.7, 115.4, 110.3, 104.4, 76.5, 61.5, 52.9, 48.6, 44.3, 35.9, 34.4, 31.8, 29.8, 27.4. HRESIMS *m*/*z* = 596.30365 [M + H]^+^ (calculated for C_37_H_43_O_2_N_3_Cl, 596.30383).

*(E)-N-((1-Benzylpiperidin-4-yl)methyl)-3-(8-methoxy-3,3-dimethyl-3,11-dihydropyrano[3,2-a]carbazol-5-yl)acrylamide***(6bg).** White solid, 64%; m.p.: 182–183 °C. ^1^H-NMR (600 MHz, DMSO-*d*_6_) δ 11.31 (s, 1H), 8.24 (s, 1H), 8.14 (s, 1H), 7.75 (d, *J* = 15.8 Hz, 1H), 7.58 (d, *J* = 2.5 Hz, 1H), 7.38–7.22 (m, 6H), 6.96 (dd, *J* = 8.6, 2.5 Hz, 1H), 6.91 (d, *J* = 9.8 Hz, 1H), 6.75 (d, *J* = 15.8 Hz, 1H), 5.84 (d, *J* = 9.8 Hz, 1H), 3.83 (s, 3H), 3.58 (s, 2H), 3.13–3.07 (m, 2H), 2.93–2.84 (m, 2H), 2.14–2.05 (m, 2H), 1.72–1.62 (m, 2H), 1.51–1.44 (m, 7H), 1.30–1.20 (m, 2H). ^13^C-NMR (150 MHz, DMSO-*d*_6_) δ 165.9, 153.5, 149.5, 138.0, 137.0, 134.9, 134.7, 129.2, 128.2, 127.3, 123.4, 123.4, 119.5, 119.2, 117.4, 117.2, 115.3, 113.8, 111.5, 104.4, 102.8, 76.5, 61.7, 55.5, 52.5, 44.0, 35.5, 29.1, 27.4. HRESIMS *m/z* = 536.29095 [M + H]^+^ (calculated for C_34_H_38_O_3_N_3_, 536.29077).

*(E)-N-((1-(2-Fluorobenzyl)piperidin-4-yl)methyl)-3-(8-methoxy-3,3-dimethyl-3,11-dihydropyrano[3,2-a]carbazol-5-yl)acrylamide***(6bh).** White solid, 56%; m.p.: 162–163 °C. ^1^H-NMR (600 MHz, DMSO-*d*_6_) δ 11.25 (s, 1H), 8.13 (s, 1H), 8.01 (d, *J* = 6.1 Hz, 1H), 7.73 (d, *J* = 15.8 Hz, 1H), 7.58 (d, *J* = 2.5 Hz, 1H), 7.42–7.36 (m, 1H), 7.33 (d, *J* = 8.6 Hz, 1H), 7.33–7.26 (m, 1H), 7.19–7.11 (m, 2H), 6.95 (dd, *J* = 8.6, 2.5 Hz, 1H), 6.90 (d, *J* = 9.8 Hz, 1H), 6.73 (d, *J* = 15.8 Hz, 1H), 5.85 (d, *J* = 9.8 Hz, 1H), 3.83 (s, 3H), 3.48 (s, 2H), 3.12–3.04 (m, 2H), 2.84–2.75 (m, 2H), 2.00–1.89 (m, 2H), 1.67–1.59 (m, 2H), 1.50–1.45 (m, 7H), 1.24–1.13 (m, 2H). ^13^C-NMR (150 MHz, DMSO-d_6_) δ 165.9, 160.8(d, *J*_C-F_ = 244.1 Hz), 153.5, 149.5, 137.9, 134.8, 134.6, 131.4 (d, *J*_C-F_ = 4.6 Hz), 129.3, 128.9 (d, *J*_C-F_ = 8.3 Hz), 125.0 (d, *J*_C-F_ = 14.6 Hz), 124.1 (d, *J*_C-F_ = 3.2 Hz), 123.4, 119.6, 119.2, 117.4, 117.1, 115.3, 115.1 (d, *J*_C-F_ = 22.1 Hz), 113.8, 111.5, 104.4, 102.8, 76.5, 59.8, 55.5, 52.8, 44.3, 35.8, 29.7, 27.4. HRESIMS *m*/*z* = 554.28119 [M + H]^+^ (calculated for C_34_H_37_O_3_N_3_F, 554.28135).

*(E)-3-(8-Methoxy-3,3-dimethyl-3,11-dihydropyrano[3,2-a]carbazol-5-yl)-N-((1-(2-methylbenzyl)piperidin-4-yl)methyl)acrylamide***(6bi).** White solid, 61%; m.p.: 181–182 °C. ^1^H-NMR (600 MHz, DMSO-*d*_6_) δ 8.14 (s, 1H), 8.02 (t, *J* = 5.8 Hz, 1H), 7.75 (d, *J* = 15.8 Hz, 1H), 7.58 (d, *J* = 2.4 Hz, 1H), 7.34 (d, *J* = 8.6 Hz, 1H), 7.22–7.18 (m, 1H), 7.16–7.07 (m, 3H), 6.96 (dd, *J* = 8.6, 2.4 Hz, 1H), 6.90 (d, *J* = 9.8 Hz, 1H), 6.74 (d, *J* = 15.8 Hz, 1H), 5.85 (d, *J* = 9.8 Hz, 1H), 3.83 (s, 3H), 3.37 (s, 2H), 3.12–3.06 (m, 2H), 2.82–2.71 (m, 2H), 2.29 (s, 3H), 1.97–1.88 (m, 2H), 1.68–1.60 (m, 2H), 1.52–1.42 (m, 7H), 1.22–1.10 (m, 2H). ^13^C-NMR (150 MHz, DMSO-d_6_) δ 165.9, 153.5, 149.5, 137.9, 136.9, 136.7, 134.8, 134.6, 130.0, 129.4, 129.3, 126.7, 125.3, 123.4, 119.6, 119.2, 117.4, 117.1, 115.3, 113.8, 111.5, 104.4, 102.8, 76.5, 60.5, 55.5, 53.2, 44.3, 36.0, 29.9, 27.4, 18.8. HRESIMS *m*/*z* = 550.30609 [M + H]^+^ (calculated for C_35_H_40_O_3_N_3_, 550.30642).

*(E)-N-((1-(2-Chlorobenzyl)piperidin-4-yl)methyl)-3-(8-methoxy-3,3-dimethyl-3,11-dihydropyrano[3,2-a]carbazol-5-yl)acrylamide***(6bj).** White solid, 63%; m.p.: 165–166 °C. ^1^H-NMR (600 MHz, DMSO-*d*_6_) δ 8.13 (s, 1H), 8.03 (d, *J* = 6.0 Hz, 1H), 7.73 (d, *J* = 15.8 Hz, 1H), 7.58 (d, *J* = 2.5 Hz, 1H), 7.52–7.23 (m, 5H), 6.95 (dd, *J* = 8.7, 2.5 Hz, 1H), 6.90 (d, *J* = 9.8 Hz, 1H), 6.73 (d, *J* = 15.8 Hz, 1H), 5.85 (d, *J* = 9.8 Hz, 1H), 3.83 (s, 3H), 3.53 (s, 2H), 3.13–3.05 (m, 2H), 2.87–2.78 (m, 2H), 2.05–1.96 (m, 2H), 1.69–1.62 (m, 2H), 1.51–1.43 (m, 7H), 1.26–1.15 (m, 2H). ^13^C-NMR (150 MHz, DMSO-d_6_) δ 165.8, 153.5, 149.5, 137.9, 136.0, 134.8, 134.6, 133.2, 130.7, 129.3, 129.2, 128.5, 127.0, 123.4, 119.6, 119.2, 117.4, 117.1, 115.3, 113.8, 111.5, 104.4, 102.8, 76.5, 58.9, 55.5, 53.1, 44.1, 35.8, 29.8, 27.4. HRESIMS *m*/*z* = 570.25189 [M + H]^+^ (calculated for C_34_H_37_O_3_N_3_Cl, 570.25180).

*(E)-N-(2-(1-Benzylpiperidin-4-yl)ethyl)-3-(8-(tert-butyl)-3,3-dimethyl-3,11-dihydropyrano[3,2-a]carbazol-5-yl)acrylamide***(6ca).** White solid, 68%. White solid, 65%; m.p.:144–145 °C. ^1^H-NMR (400 MHz, DMSO-*d*_6_) δ 11.34 (s, 1H), 8.17 (s, 1H), 8.01 (d, *J* = 1.8 Hz, 1H), 7.97 (t, *J* = 5.6 Hz, 1H), 7.74 (d, *J* = 15.8 Hz, 1H), 7.40 (dd, *J* = 8.5, 1.9 Hz, 1H), 7.37–7.31 (m, 5H), 7.30–7.26 (m, 1H), 6.93 (d, *J* = 9.8 Hz, 1H), 6.71 (d, *J* = 15.8 Hz, 1H), 5.85 (d, *J* = 9.8 Hz, 1H), 3.60 (s, 2H), 3.27–3.14 (m, 2H), 2.89 (s, 2H), 2.26–1.98 (m, 2H), 1.75–1.64 (m, 2H), 1.48 (s, 6H), 1.45–1.40 (m, 2H), 1.38 (s, 9H), 1.29–1.13 (m, 3H). ^13^C-NMR (100 MHz, DMSO-*d*_6_) δ 165.7, 149.3, 143.9, 141.7, 138.4, 137.8, 137.7, 134.4, 129.3, 128.9, 128.2, 122.8, 122.6, 119.6, 118.7, 117.5, 117.3, 115.7, 115.3, 110.3, 104.4, 76.5, 59.8, 52.8, 36.1, 35.9, 34.4, 32.3, 31.8, 31.2, 27.4. HRESIMS *m/z* = 576.35858 [M + H]^+^ (calculated for C_38_H_46_O_2_N_3_, 576.35845).

*(E)-3-(8-(tert-Butyl)-3,3-dimethyl-3,11-dihydropyrano[3,2-a]carbazol-5-yl)-N-(2-(1-(2-fluorobenzyl)piperidin-4-yl)ethyl)acrylamide***(6cb).** White solid, 57%; m.p.: 122–123 °C. ^1^H-NMR (400 MHz, DMSO-*d*_6_) δ 11.35 (s, 1H), 8.17 (s, 1H), 8.01 (s, 1H), 7.96 (t, *J* = 5.7 Hz, 1H), 7.73 (d, *J* = 15.8 Hz, 1H), 7.40 (dd, *J* = 8.4, 1.8 Hz, 1H), 7.38–7.26 (m, 3H), 7.19–7.11 (m, 2H), 6.93 (d, *J* = 9.8 Hz, 1H), 6.71 (d, *J* = 15.8 Hz, 1H), 5.85 (d, *J* = 9.8 Hz, 1H), 3.49 (s, 2H), 3.27–3.15 (m, 2H), 2.83–2.76 (m, 2H), 2.02–1.88 (m, 2H), 1.71–1.59 (m, 2H), 1.48 (s, 6H), 1.45–1.39 (m, 2H), 1.38 (s, 10H), 1.21–1.08 (m, 2H).^13^C-NMR (100 MHz, DMSO-d_6_) δ 165.7, 160.8 (d, *J*_C-F_ = 244.5 Hz), 149.3, 141.8, 138.4, 137.8, 134.3, 131.5 (d, *J*_C-F_ = 5.0 Hz), 129.3, 128.9 (d, *J*_C-F_ = 8.3 Hz), 124.9 (d, *J*_C-F_ = 14.6 Hz), 124.1 (d, *J*_C-F_ = 3.2 Hz), 122.8, 122.6, 119.6, 118.7, 117.5, 117.4, 115.7, 115.3, 115.1 (d, *J*_C-F_ = 22.0 Hz), 110.3, 104.4, 76.5, 54.9, 53.1, 36.2, 36.0, 35.0, 34.4, 32.7, 31.8, 27.4. HRESIMS *m*/*z* = 594.34808 [M + H]^+^ (calculated for C_38_H_45_O_2_N_3_F, 594.34903).

*(E)-3-(8-(tert-Butyl)-3,3-dimethyl-3,11-dihydropyrano[3,2-a]carbazol-5-yl)-N-(2-(1-(2-methylbenzyl)piperidin-4-yl)ethyl)acrylamide***(6cc)**. White solid, 48%; m.p.: 158–159 °C. ^1^H-NMR (400 MHz, DMSO-*d*_6_) δ 11.34 (s, 1H), 8.17 (s, 1H), 8.02 (d, *J* = 1.8 Hz, 1H), 7.96 (t, *J* = 5.6 Hz, 1H), 7.75 (d, *J* = 15.7 Hz, 1H), 7.40 (dd, *J* = 8.6, 1.8 Hz, 1H), 7.35 (d, *J* = 8.6 Hz, 1H), 7.22–7.16 (m, 1H), 7.14–7.08 (m, 3H), 6.93 (d, *J* = 9.8 Hz, 1H), 6.72 (d, *J* = 15.7 Hz, 1H), 5.85 (d, *J* = 9.8 Hz, 1H), 3.36 (s, 2H), 3.26–3.17 (m, 2H), 2.81–2.71 (m, 2H), 2.29 (s, 3H), 1.96–1.86 (m, 2H), 1.69–1.60 (m, 2H), 1.48 (s, 6H), 1.45–1.39 (m, 2H), 1.39–1.37 (m, 10H), 1.19–1.05 (m, 2H). ^13^C-NMR (100 MHz, DMSO-d_6_) δ 165.7, 149.3, 141.8, 138.4, 137.8, 136.9, 136.8, 134.4, 130.0, 129.4, 129.3, 126.7, 125.3, 122.8, 122.6, 119.6, 118.7, 117.5, 117.4, 115.7, 115.4, 110.3, 104.4, 76.5, 60.6, 53.5, 36.3, 36.1, 34.4, 32.9, 32.0, 31.8, 27.4, 18.8. HRESIMS *m*/*z* = 590.37402 [M + H]^+^ (calculated for C_39_H_48_O_2_N_3_, 590.37410).

*(E)-3-(8-(tert-Butyl)-3,3-dimethyl-3,11-dihydropyrano[3,2-a]carbazol-5-yl)-N-(2-(1-(2-chlorobenzyl)piperidin-4-yl)ethyl)acrylamide***(6cd).** White solid, 56%; m.p.: 160–161 °C. ^1^H-NMR (400 MHz, DMSO-*d*_6_) δ 11.31 (s, 1H), 8.17 (s, 1H), 8.02 (d, *J* = 1.8 Hz, 1H), 7.96 (t, *J* = 5.6 Hz, 1H), 7.75 (d, *J* = 15.8 Hz, 1H), 7.49–7.23 (m, 6H), 6.92 (d, *J* = 9.8 Hz, 1H), 6.72 (d, *J* = 15.8 Hz, 1H), 5.85 (d, *J* = 9.8 Hz, 1H), 3.51 (s, 2H), 3.26–3.17 (m, 2H), 2.86–2.71 (m, 2H), 2.08–1.92 (m, 2H), 1.72–1.58 (m, 2H), 1.48 (s, 6H), 1.45–1.40 (m, 2H), 1.38 (s, 10H), 1.27–1.05 (m, 2H). ^13^C-NMR (100 MHz, DMSO-d_6_) δ 165.7, 149.3, 141.8, 138.4, 137.8, 136.1, 134.4, 133.2, 130.7, 129.3, 129.2, 128.4, 126.9, 122.8, 122.6, 119.6, 118.7, 117.5, 117.4, 115.7, 115.4, 110.3, 104.4, 76.4, 59.0, 53.4, 36.9, 36.2, 34.9, 34.4, 32.7, 31.8, 27.4. HRESIMS *m*/*z* = 610.31873 [M + H]^+^ (calculated for C_38_H_45_O_2_N_3_Cl, 610.31948).

*(E)-N-(2-(1-Benzylpiperidin-4-yl)ethyl)-3-(8-methoxy-3,3-dimethyl-3,11-dihydropyrano[3,2-a]carbazol-5-yl)acrylamide***(6ce).** White solid, 47%; m.p.:152–153 °C. ^1^H-NMR (600 MHz, DMSO-*d*_6_) δ 8.14 (s, 1H), 7.98 (t, *J* = 5.6 Hz, 1H), 7.73 (d, *J* = 15.8 Hz, 1H), 7.59 (d, *J* = 2.5 Hz, 1H), 7.38–7.18 (m, 6H), 6.96 (dd, *J* = 8.7, 2.5 Hz, 1H), 6.90 (d, *J* = 9.8 Hz, 1H), 6.69 (d, *J* = 15.8 Hz, 1H), 5.85 (d, *J* = 9.8 Hz, 1H), 3.83 (s, 3H), 3.44 (s, 2H), 3.25–3.16 (m, 2H), 2.86–2.73 (m, 2H), 1.98–1.84 (m, 2H), 1.71–1.58 (m, 2H), 1.48 (s, 6H), 1.44–1.37 (m, 2H), 1.36–1.25 (m, 1H), 1.22–1.09 (m, 2H). ^13^C-NMR (150 MHz, DMSO-*d*_6_) δ 165.7, 153.5, 149.5, 138.1, 137.9, 134.8, 134.6, 129.3, 128.8, 128.1, 126.8, 123.4, 119.7, 119.2, 117.4, 117.1, 115.3, 113.8, 111.5, 104.4, 102.8, 76.5, 62.4, 55.5, 53.2, 36.3, 36.0, 32.8, 31.8, 27.4. HRESIMS *m/z* = 550.30670 [M + H]^+^ (calculated for C_35_H_40_O_3_N_3_, 550.30642).

*(E)-N-(2-(1-(2-Fluorobenzyl)piperidin-4-yl)ethyl)-3-(8-methoxy-3,3-dimethyl-3,11-dihydropyrano[3,2-a]carbazol-5-yl)acrylamide***(6cf).** White solid, 58%; m.p.: 141–142 °C. ^1^H-NMR (400 MHz, DMSO-*d*_6_) δ 11.26 (s, 1H), 8.13 (s, 1H), 7.96 (t, *J* = 5.6 Hz, 1H), 7.72 (d, *J* = 15.8 Hz, 1H), 7.58 (d, *J* = 2.5 Hz, 1H), 7.41–7.36 (m, 1H), 7.33 (d, *J* = 8.7 Hz, 1H), 7.31–7.26 (m, 1H), 7.19–7.11 (m, 2H), 6.95 (dd, *J* = 8.7, 2.5 Hz, 1H), 6.90 (d, *J* = 9.9 Hz, 1H), 6.68 (d, *J* = 15.8 Hz, 1H), 5.85 (d, *J* = 9.9 Hz, 1H), 3.83 (s, 3H), 3.48 (s, 2H), 3.24–3.15 (m, 2H), 2.84–2.72 (m, 2H), 1.98–1.88 (m, 2H), 1.70–1.59 (m, 2H), 1.48 (s, 6H), 1.44–1.35 (m, 2H), 1.25–1.21 (m, 1H), 1.20–1.08 (m, 2H). ^13^C-NMR (100 MHz, DMSO-d_6_) δ 165.7, 160.8 (d, *J*_C-F_ = 244.2 Hz), 153.5, 149.4, 138.0, 134.9, 134.5, 131.5 (d, *J*_C-F_ = 4.5 Hz), 129.3, 128.9 (d, *J*_C-F_ = 8.1 Hz), 124.9 (d, *J*_C-F_ = 14.7 Hz), 124.1(d, *J*_C-F_ = 3.5 Hz), 123.4, 119.7, 119.2, 117.4, 117.2, 115.3, 115.1(d, *J*_C-F_ = 22.0 Hz), 113.8, 111.5, 104.4, 102.8, 76.5, 55.5, 54.9, 53.1, 36.2, 36.0, 32.7, 31.8, 27.4. HRESIMS *m*/*z* = 568.29767 [M + H]^+^ (calculated for C_35_H_39_O_3_N_3_F, 568.29809).

*(E)-3-(8-Methoxy-3,3-dimethyl-3,11-dihydropyrano[3,2-a]carbazol-5-yl)-N-(2-(1-(2-methylbenzyl)piperidin-4-yl)ethyl)acrylamide***(6cg)**. White solid, 51%; m.p.: 161–162 °C. ^1^H-NMR (600 MHz, DMSO-*d*_6_) δ 8.14 (s, 1H), 7.98 (t, *J* = 5.7 Hz, 1H), 7.74 (d, *J* = 15.8 Hz, 1H), 7.59 (d, *J* = 2.5 Hz, 1H), 7.34 (d, *J* = 8.6 Hz, 1H), 7.22–7.07 (m, 4H), 6.96 (dd, *J* = 8.6, 2.5 Hz, 1H), 6.90 (d, *J* = 9.8 Hz, 1H), 6.70 (d, *J* = 15.8 Hz, 1H), 5.85 (d, *J* = 9.8 Hz, 1H), 3.83 (s, 3H), 3.37 (s, 2H), 3.26–3.18 (m, 2H), 2.80–2.71 (m, 2H), 2.29 (s, 3H), 1.95–1.85 (m, 2H), 1.68–1.60 (m, 2H), 1.48 (s, 6H), 1.42–1.37 (m, 2H), 1.35–1.27 (m, 1H), 1.20–1.06 (m, 2H). ^13^C-NMR (150 MHz, DMSO-d_6_) δ 165.7, 153.5, 149.5, 137.9, 136.9, 136.7, 134.8, 134.6, 130.0, 129.4, 129.3, 126.7, 125.3, 123.4, 119.7, 119.2, 117.4, 117.1, 115.3, 113.8, 111.5, 104.4, 102.8, 76.5, 60.5, 55.5, 53.5, 36.3, 36.0, 33.0, 32.0, 27.4, 18.8. HRESIMS *m*/*z* = 564.32147 [M + H]^+^ (calculated for C_36_H_42_O_3_N_3_,564.32207).

*(E)-N-(2-(1-(2-Chlorobenzyl)piperidin-4-yl)ethyl)-3-(8-methoxy-3,3-dimethyl-3,11-dihydropyrano[3,2-a]carbazol-5-yl)acrylamide***(6ch).** White solid, 57%; m.p.: 145–146 °C. ^1^H-NMR (400 MHz, DMSO-*d*_6_) δ 11.24 (s, 1H), 8.13 (s, 1H), 7.97 (t, *J* = 5.6 Hz, 1H), 7.72 (d, *J* = 15.8 Hz, 1H), 7.59 (d, *J* = 2.5 Hz, 1H), 7.51–7.38 (m, 2H), 7.36–7.23 (m, 3H), 6.95 (dd, *J* = 8.7, 2.5 Hz, 1H), 6.90 (d, *J* = 9.8 Hz, 1H), 6.69 (d, *J* = 15.8 Hz, 1H), 5.85 (d, *J* = 9.8 Hz, 1H), 3.83 (s, 3H), 3.53 (s, 2H), 3.27–3.17 (m, 2H), 2.91–2.74 (m, 2H), 2.07–1.93 (m, 2H), 1.72–1.62 (m, 2H), 1.48 (s, 6H), 1.45–1.37 (m, 2H), 1.38–1.26 (m, 1H), 1.23–1.11 (m, 2H). ^13^C-NMR (100 MHz, DMSO-d6) δ 165.7, 153.5, 149.4, 138.0, 136.1, 134.9, 134.5, 133.2, 130.7, 129.3, 129.2, 128.5, 127.0, 123.4, 119.7, 119.2, 117.4, 117.2, 115.3, 113.8, 111.5, 104.4, 102.8, 76.5, 59.0, 55.5, 53.4, 36.2, 36.0, 32.7, 31.9, 27.4. HRESIMS *m/z* = 584.26843 [M + H]^+^ (calculated for C_35_H_39_O_3_N_3_Cl, 584.26745).

### 3.2. Biological Evaluation

#### 3.2.1. Anti-Cholinesterase Activity Assays

The activity of AChE was tested using an acetylcholinesterase activity detection kit (A024-1-1, Jian Cheng, Nanjing). AChE is a kind of serine hydrolase, which exists widely in the tissue and serum of animals. AChE acts as a key role in nervous conduction. It can hydrolyze acetyl choline into acetyl coenzyme A and choline. The activity of BuChE was tested using a butyrylcholinesterase activity detection kit (A025-1-1, Jian Cheng, Nanjing). It can hydrolyze butyryl choline into acetic acid and choline. Choline can interact with disulfide *p*-nitrobenzoic acid (DTNB) to produce 5-mercapto-nitrobenzoic acid (TNB), which has an absorption peak at 412 nm. Thus, we could analyze the increasing rate of 412 nm absorption to calculate the activity of AChE and BuChE.

The specific analysis method was as follows: a standard solution of AChE (Sigma-Aldrich) was prepared in 50 mM Tris-HCl, pH = 8.0, 0.1% *w*/*v*, bovine serum albumin (BSA) into 0.22 U/mL, and a standard solution of BuChE (Sigma-Aldrich) was prepared in 50 mM Tris-HCl, pH = 8.0, 0.1% *w*/*v*, BSA into 0.12 U/mL. Next, 50 μL of the above standard solution and 10 μL of different concentrations of drug solution (0, 20 μM, 40 μM, 60 μM, 80 μM, and 100 μM) were used to test the change in activity. The procedure was processed as described by the kit instructions.

#### 3.2.2. Primary Cortical Neuron Culture and Treatment

Rat cortical neurons were prepared from newborn Sprague-Dawley (SD) rats. All cell suspensions were grown in neurobasal/B27 medium with 100-U/mL penicillin and 100-μg/mL streptomycin. The cells were plated in poly-L-lysine (PLL, 0.1 mg/mL)-coated multiwell plates or chamber slides and maintained at 37 °C in a humidified atmosphere of 5% CO_2_ and 95% air. Seven day cultures were used for treatment. The treatment methods were similar to those used for PC12, but the concentrations of H_2_O_2_ and sodium dithionite were 100 μM and 2 mM, respectively.

#### 3.2.3. Cell Viability

Cell viability was determined by MTT assay. Cells were incubated with 5 mg/mL MTT (10 μL) at 37 °C for 4 h. The formazan crystals were dissolved, and absorbance was determined at 570 nm using a Microplate Reader (Thermo, Germany). The cell viability was expressed as a percentage of the optical density (OD) value of the control cultures.

#### 3.2.4. Parallel Artificial Membrane Permeability Assay

The compounds’ blood–brain barrier penetration was evaluated by using the parallel artificial membrane permeation assay (PAMPA) described by Di et al. [20]. The two commercial drugs were purchased from Aladdin (China). Porcine brain lipid was obtained from Sigma (China). The donor microplate (polyvinylidene fluoride (PVDF) membrane, pore size 0.45 *μ*m) and acceptor microplate were both from Millipore (China). The 96-well ultraviolet (UV) plate (COSTAR) was from Corning Incorporated (USA). The detailed procedure was described in [26,27].

#### 3.2.5. Free-Radical Scavenging

ESR signals were recorded with a 10 mW incident microwave and 10,000 MHz field modulation of 1 G. The scan width was 150 G for 2, 2, 6, 6-tetramethylpiperidine (TEMP) experiments. All measurements were performed at room temperature (RT). Briefly, 5 × 10^−2^ mol/L TEMP and 10^−2^ mol/L **6bd** or Eda were mixed in DMSO, and ESR signals were obtained after exposure to UVA light for 1 min.

## 4. Conclusions

In our previous investigations, we identified several carbazole alkaloids with neuroprotective effects, such as Clau F and CZ-7. In this study, a series of Claulansine F–donepezil hybrids were designed and synthesized as multitarget drugs and their cholinesterase inhibitory activities were evaluated. On this basis, we discussed the SAR. Among the tested compounds, the compounds with R= t-Bu exhibited the better potency against cholinesterase. Compound **6bd** showed the strongest neuroprotective effects in vitro. Furthermore, **6bd** could cross the blood–brain barrier in vitro. Most importantly, **6bd** showed stronger free-radical scavenging capacity than Eda. Therefore, **6bd** may be valuable for intervention against Alzheimer’s disease. We are interested in further investigating the mechanisms underlying the inhibition of Alzheimer’s disease by **6bd**.

## Data Availability

The data presented in this study are available on request from the corresponding author.

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
