# Peer review of "Claulansine F–Donepezil Hybrids as Anti-Alzheimer’s Disease Agents with Cholinergic, Free-Radical Scavenging, and Neuroprotective Activities"

_molecules, 2021, doi:10.3390/molecules26051303_

Round 1
Reviewer 1 Report
The work is interesting, however, several issues should be corrected before the publishing.
In abstract abbreviated compounds should be cited as chemical entities.
In Lines 29 and 38 more recent reviews about progress in Alzheimer's disease investigations should be cited.
Language corrections should be performed:
e.g. Line 76 - were synthesized,
Line 115 - retain activity,
Line 123 - candidate compound,
Line 134 - according to the obtained data on permeability
Author Response
Thank you for your letter and for the reviewer’s comments. As the reviewers’ suggests, The following revisions have been made: (1) In abstract, “the compound 6bd” was changed into “(E)-3-(8-(tert-butyl)-3,3-dimethyl-3,11-dihydropyrano[3,2-a]carbazol-5-yl)-N-((1-(2-chlorobenzyl)piperidin-4-yl)methyl)acrylamide (6bd)”; (2) We added 3 recent reviews published in the journal of molecules about progress in Alzheimer's disease investigations. (3) We checked English usage and typographical mistakes carefully and 4 language errors put forward by reviewers were corrected.
Reviewer 2 Report
This manuscript describes an interesting series of pyranocarbazole/benzylamine conjugates, and their biological testing. The synthetic methods are conventional but effective, and the NMR proof of identity is first rate. I especially like that H-F coupling constants are reported. Unfortunately, other aspects of characterization are less satisfactory. (1) Every compound is described as a “white solid”, but there are no melting points, at all. (2) There are no microanalyses either. Therefore proof of purity must rest on the NMR traces, but (3) about half the 1H NMR traces show small amounts of ethyl acetate. If the characterization were cleaned up, I would support publication.
Author Response
Thank you for your letter and for the reviewer’s comments. As the reviewers’ suggests, The following revisions have been made: (1) We added the data of melting points in the manuscript. (2) We added HPLC analysis of all final compounds in the supplementary document. (3) We have redone the H NMR of compounds 6ab, 6ad, 6af, 6ag, 6ah, 6bh, 6bi, 6ca, 6ce and 6cg and the characterization of ethyl acetate were cleaned up.
Round 2
Reviewer 2 Report
My criticisms have been adequately addressed, I recommend publication.